# Bulk transparent supramolecular glass enabled by host–guest molecular recognition

Changyong Cai[1], Shuanggen Wu[1], Yunfei Zhang[1], Fenfang Li[2], Zhijian Tan ®[3] ✉ & Shengyi Dong ®[1] ✉

Supramolecular glass is a non-covalently cross-linked amorphous material that exhibits excellent optical properties and unique intrinsic structural features. Compared with artificial inorganic/organic glass, which has been extensively developed, supramolecular glass is still in the infancy stage, and itself is rarely recognized and studied thus far. Herein, we present the development of the host–guest molecular recognition motifs between methyl-β-cyclodextrin and *para*-hydroxybenzoic acid as the building blocks of supramolecular glass. Non-covalent polymerization resulting from the host–guest complexation and hydrogen bonding formation enables high transparency and bulk state to supramolecular glass. Various advantages, including recyclability, compatibility, and thermal processability, are associated with dynamic assembly pattern. Short-range order (host–guest complexation) and long-range disorder (three dimensional polymeric network) structures are identified simultaneously, thus demonstrating the typical structural characteristics of glass. This work provides a supramolecular strategy for constructing transparent materials from organic components.

The development of transparent materials is essential for industrial production and scientific activity[1–6]. In ancient times, gemstone, crystals, and amber were frequently used as natural transparent materials to fabricate optical devices[7–9]. Since the earliest appearance of man-made glass, commercially available inorganic glass has substantially contributed to the rapid growth of transparent materials[10–12]. The development of artificial glass has been regarded as a significant milestone in the long history of transparent materials. With advancements in polymer science, organic glass, which originates from covalently linked polymers, has become indispensable for various applications[13–15]. Currently, inorganic and organic materials are two classes of the most extensively used transparent materials[16].

Following the considerable success of organic glass, noncovalent interactions, and molecular recognition motifs have been used to fabricate transparent materials, with gel as a typical representative[17–22].

Supramolecular gels with high transmittance values have various dynamic features and can be used for diverse applications[23–27]. Compared with organic glass, supramolecular gels are predominantly soft and ductile, indicating that these gels are not viable alternatives to modern glass materials[28–31]. In principle, supramolecular glass can be assembled from organic components via non-covalent bonding, similar to its gel counterpart[32–36]. However, supramolecular glass has not been sufficiently recognized and investigated thus far, especially for its intrinsic structure, driving force, and mechanical properties[16,37,38].

The selection of suitable building blocks is critical for the formation of transparent supramolecular glass. Although, macrocycles and the related recognition motifs have been widely used, the application of macrocycles and host–guest complexes have not been the first choice for glass formation[23,39,40]. In this study, the molecular recognition motif of methyl-β-cyclodextrin [**M**] and *para*-hydroxybenzoic acid

[1]College of Chemistry and Chemical Engineering, Hunan University, Changsha, Hunan 410082, P. R. China. [2]College of Chemistry and Chemical Engineering, Central South University, Changsha, Hunan 410083, P. R. China. [3]Institute of Bast Fiber Crops, Chinese Academy of Agricultural Sciences, Changsha, Hunan 410205, P. R. China. ✉e-mail: tanzhijian@caas.cn; dongsy@hnu.edu.cn

[**H**] is used as the basic unit to construct supramolecular glass **MH**. The recognition behavior between **M** and **H** yields the order host–guest recognition structures on a restricted scale and actuates the supramolecular polymerization of **M/H** host–guest complexes into isotropic **MH** in bulk. The nature and intrinsic structure of **MH** were carefully studied via a combined experimental and theoretical investigation. The obtained **MH** exhibits various excellent performances in terms of high transmittance, good thermal processability, broad compatibility, and considerable recyclability.

## Results

### Preparation and structural analysis of supramolecular glass

To obtain transparent and smooth supramolecular glass in bulk, the process of synthesizing **MH** was divided into two steps. Initially, a supramolecular polymer was prepared by evaporating an aqueous mixture of **M** and **H** at 80 °C. Subsequently, the newly formed crude material was hot-pressed for 10 min, while maintaining the temperature and pressure at 80 °C and 20 MPa, respectively; thus, transparent and bulk supramolecular glass **MH** was constructed (Fig. 1a and Supplementary Tab. 1)[40–42]. Different molar ratios of **M** and **H** were tried, and only a limited ratios of **M** and **H** is available for glass formation (Supplementary Tab. 2).

According to the abovementioned information, **MH** formation involves a transition from an aqueous solution to a bulk state. Therefore, supramolecular glass was initially studied in a solution state. As shown in the nuclear magnetic resonance ($^1$H NMR) spectra, the signals of aromatic protons on **H** show down-field shifts, from 6.76 ($H_b$) and 7.73 ($H_a$) ppm to 6.90 and 7.84 ppm, respectively, along with the

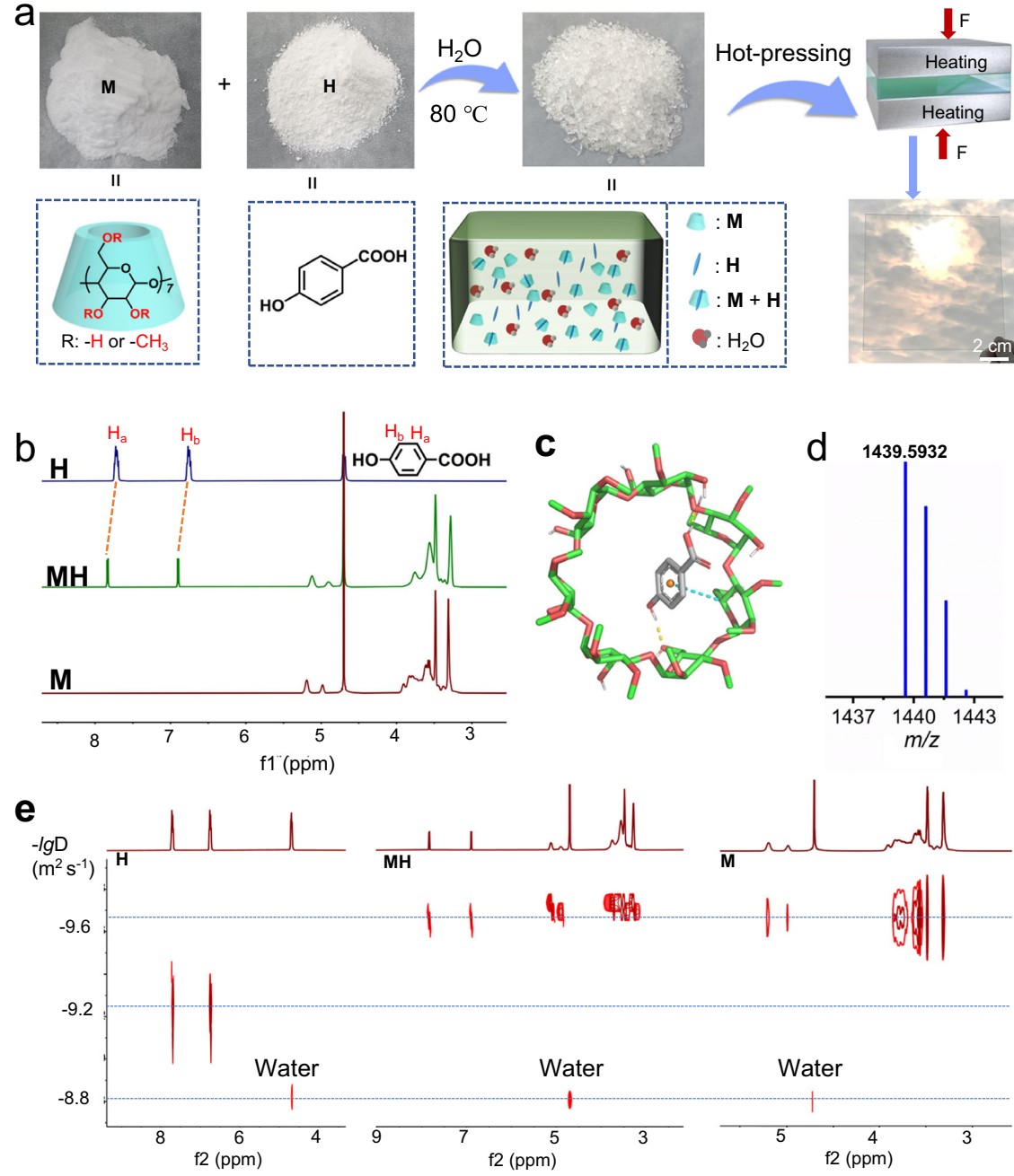

**Fig. 1 | Preparation and characterization of MH. a** Preparation process and possible assembly motifs of **MH** glass. **b** $^1$H NMR spectra of **M, H** and **MH** (400 MHz, D$_2$O, 50 mg mL$^{-1}$, 25 °C). **c** Model of **M** and **H** host–guest complex. **d** ESI-MS spectrum of **M** and **H** ($m/z$: 1439.5932). **e** DOSY spectra (400 MHz, D$_2$O, 15.3 mM, 25 °C) of **M, H** and **MH**.

changes in their peak shapes (Fig. 1b and Supplementary Fig. 1). Those NMR results suggested the aromatic unit of **H** is not surrounded by solvent molecules, but shielded by nonaromatic **M** (Fig. 1b)[43]. The electron spray ionization mass (ESI–MS) data further supports the binding behavior between **M** and **H** in the solution (Fig. 1d), because a peak at 1439.5932 (**M@H**) was found in its high-resolution mass spectrum.

More decisive evidence of the complexation between **M** and **H** was obtained from 2D diffusion-ordered spectroscopy (DOSY) spectra. Only one set of diffusion coefficients ($D_H$) was recorded for **M/H** complexes (Fig. 1e). Additionally, the diffusion coefficient of **M/H** ($1.82 \times 10^{-10}$ m$^2$ s$^{-1}$) is approximately identical to that of the individual **M** ($1.66 \times 10^{-10}$ m$^2$ s$^{-1}$) at the same concentration (15.3 mM), but much lower than that of **H** ($7.59 \times 10^{-10}$ m$^2$ s$^{-1}$), when the residual water was used as a reference ($1.48 \times 10^{-9}$ m$^2$ s$^{-1}$). These observations confirm the molecular recognition between **M** and **H** and also indicate that **H** is located in the cavity of **M** to form a threaded structure. Because the molecular configurations of **M** and threaded **M/H** are similar, they demonstrate similar diffusion behaviors in D$_2$O[44]. Simulation results of the possible threaded **M/H** molecular pattern are consistent with the structural information obtained from NMR characterization (Fig. 1c and Supplementary Fig. 2). These results demonstrate that **M** and **H** form host–guest structures in solution.

During glass formation, numerous water molecules were removed via evaporation. Thus, although **M** and **H** favor to form threaded structures via host–guest molecular recognition in solution, the thermal evaporation and rapid annealing behavior during the preparation result in multiple recognition patterns of **M** and **H** in the bulk state[39]. Two-dimensional correlation analysis of the Fourier-Transform Infrared (FT-IR) spectra was performed according to temperature-dependent IR (20–120 °C) (Supplementary Figs. 3–5). A negative cross peak is observed at 3404 and 3730 cm$^{-1}$, suggesting the coexistence of threaded and non-threaded **M/H** complexes; the threaded complex demonstrates a higher thermostability. In the control experiments, supramolecular glass was successfully cast from **M** and **H** mixtures at molar ratios of 1:2 or 2:1 (Supplementary Tab. 2), indicating that the threaded **MH** complex is not the only building block in supramolecular glass[40]. The simulated binding patterns support the possibility of different **M/H** recognition motifs, such as semi-threaded and non-threaded complexation structures, thereby displaying the structural diversity of building blocks in supramolecular glass. These **MH** motifs exhibit stable binding patterns and high binding affinities ($E_b$: $-6.97\sim -24.09$ kcal mol$^{-1}$), as shown in Figs. 2a–d.

## Structural water in supramolecular glass

Solvent-evaporation was used in the preparation of supramolecular glass, therefore it is reasonable that there are water molecules remained in the bulk materials. However, the role and importance of water in the glass formation is still ambiguous[35,36,38]. In this study, according to the thermogravimetric analysis (TGA) result of **MH**, it was observed that supramolecular glass contains approximately 4.0 wt% water (Supplementary Fig. 6). Therefore, the mode of the existence of water in **MH** must be elucidated. In our previous study[45–47], it has been found that a small number of water molecules serve as essential monomers in supramolecular polymerization. These water molecules bridge immiscible clusters to yield an isotropic phase through hydrogen bonding formation and are referred to as "structural water" to distinguish them from solvent water molecules[45,48]. In the following investigation, attentions were focused on structural water in **MH**.

Broadband dielectric spectroscopy experiments were performed to investigate the water molecules in **MH**. As shown in Fig. 3a and Supplementary Fig. 7, no discontinuity is observed in the plots of σ$_{DC}$ over $1/T$ for **MH** around 0 °C (the freezing point of water in bulk), indicating that water molecules in **MH** may exist in an inseparable state, and the formation of bulk water clusters in glass formation is

unfavorable. The crystal structure of **H** demonstrates that water molecules can form multiple hydrogen bonds with **H** (Supplementary Fig. 8). The simulated results also show that **M** and **H** can be bonded with water molecules through flexible binding patterns (Fig. 3b). Most of the simulated **M**/water or **H**/water structures have moderate to high binding affinities, with the binding energies from $-1.14$ to $-58.38$ kcal mol$^{-1}$. Low-field NMR spectrum provides quantitative results and further support the existence of structural water, because around 98.4% of water in **MH** is firmly bound water with the relaxation time of 0.1 ms (Supplementary Fig. 9). These observations demonstrate that water molecules play an essential role in the formation of supramolecular polymeric structure.

The above results demonstrate that the structural water molecules in **MH** facilitate the glass formation: different **M/H** recognition motifs can self-assemble into three-dimensional networks; the water molecules involved provide additional hydrogen bonding sites and increase the cross-linking density of supramolecular glass[39,49]. The water molecules effectively increase the cohesive energy density of **MH** from $4.70 \times 10^8$ to $5.20 \times 10^8$ J m$^{-3}$, indicating that structural water contributes to a higher intrinsic interaction intensity (Fig. 3d, Supplementary Tab. 5).

Water significantly influences the structural configuration of **MH**. In general, water molecules reduce the free volume of supramolecular glass (Fig. 3c, Supplementary Figs. 10,11 and Supplementary Tabs. 3, 4). For example, the fraction of free volume (FFV) of water-free **MH** (**M**:**H** at 20:20) is 19.13%, which is higher than that of **MH** with a molar ratio of **M**: **H**: water of 20:20:5 (14.75%). This observation indicates that supramolecular glass with water molecules has a more compact polymerization pattern and higher intrinsic interaction intensity due to the existence of water and water-involved cross-linking behavior. Meanwhile, compared with "free water", structural water is more stable (Supplementary Fig. 6), because the TGA result of the freshly prepared **MH** is almost the same to that of long-term stored **MH** (300 days).

## The interior structure of supramolecular glass

In the preparation of **MH**, a transition from a diluted solution to a solid glass was observed, during which water molecules were removed, and the viscosity of glass increased rapidly (Supplementary Fig. 12). Those observations may indicate that glass is in a thermodynamically meta-stable state, and the movements of dynamic recognition motifs are suppressed, due to tits high viscosity. Thus, attentions were focused on the investigation of the interior structure of **MH**. It was found that **MH** is amorphous (Supplementary Figs. 13,14), because only broad peaks at around 10 and 20 degrees were observed from its powder X-ray diffraction spectra[50]. Meanwhile, no peaks were found in the small angle X-ray scattering spectra (Supplementary Fig. 15). The radial distribution function (RDF) of **MH** was further performed to study its structural characteristic. RDF results demonstrate that **MH** exhibits a distinctive peak at approximately 1.93 Å, indicating the existence of the relatively order structure in supramolecular glass (Fig. 3e)[51,52]. Water molecules are independent of the short-range order structure, because the same peak at 1.93 Å is observed in the RDF spectra of the water-free **MH** system. These results suggest that the short-range order structure is most likely related to the recognition motifs of **M/H**.

The centroid distances of **M/H** were subsequently calculated, with the threaded **M/H** motif presenting the shortest centroid distance of 2.0 Å. The semi-threaded **M/H** structures have centroid distances of 2.0–6.0 Å (Fig. 3f). The distribution ratios of **M/H** complexes in supramolecular glass were calculated to be 15, 40, and 45% for threaded **MH**, multiple semi-threaded **MH**, and uncomplexed **M** and **H**, respectively. Based on these findings, versatile host–guest complexes can be reasonably considered as the source of the short-range order structure. Furthermore, it is concluded that supramolecular glass formation is the fruit of different building blocks, which was rarely discussed in supramolecular glass systems[35,36,38].

## Properties of supramolecular glass

After hot-pressing and cooling, the newly formed **MH** (size: >10 cm; thickness: <1.0 mm) was colorless and transparent. As shown in Fig. 4a, a thin **MH** plate cannot be easily identified with the naked eye when it is placed over a colorful painting. Based on quantitative tests, the excellent transparency (>85%) of **MH** over a wide wavelength range

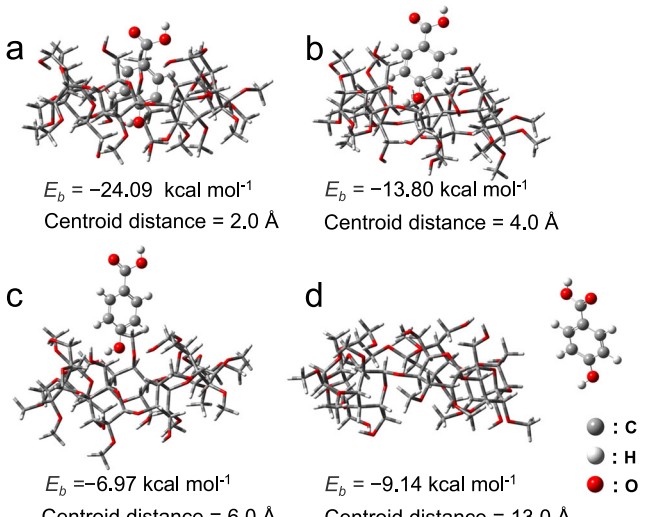

**Fig. 2 | Model of M and H based on different binding energy ($E_b$) and centroid distance. a** Centroid distance of 2 Å. **b** Centroid distance of 4 Å. **c** Centroid distance of 6 Å. **d** Centroid distance of 13 Å.

(300 to 1000 nm) is comparable to that of commercially available glass materials and some reported supramolecular glasses (Fig. 4b)[35,36]. The solid-phase ultraviolet spectrum of **MH** shows weak absorptions in the visible and near-infrared regions, demonstrating **MH** is transparent in the visible and near-infrared regions (Supplementary Fig. 16). In contrast, after drying under vacuum, **MH** becomes an opaque material, and its visible-light transmittance is below 10% (Figs. 4b, c and Supplementary Fig. 18), directly showing the great importance of water in the optical properties of supramolecular glass. By comparing the NMR spectra of **MH** and dried **MH**, it was observed that there are no covalent reactions between **M** and **H** in dried **MH**. Because the two [1]H NMR spectra are almost the same (Supplementary Fig. 1). Dried **MH** and **MH** with different water contents are amorphous due to the absence of crystalline peaks in their PXRD spectra (Supplementary Fig. 14)[50]. Meanwhile, dried **MH** has a similar DSC curve of **MH** with water (Supplementary Fig. 17). A possible explanation of the transition from transparency to opacity of **MH** is that structural water is replaced by air, which leads to the light scattering performance[46]. In addition, the refractive indexes of **MH** are between 1.55 and 1.46 when the wavelength ranges from 250 to 2000 nm (Supplementary Fig. 21). The refractive index of **MH** approaches that of common optical glass (-1.5)[53].

The Brunner−Emmet−Teller surface area of **MH** is measured to be only 0.49 m$^2$ g$^{-1}$, which suggests that **MH** has a compact and non-porous structure[54]. In addition to other techniques, atomic force microscopy (AFM) characterization was selected to investigated the microscopic mechanical properties of supramolecular glass (Figs. 4d, e). A smooth surface morphology was recorded for **MH**, with the root-mean-square roughness below 1.61 nm, which can be attributed to the hot-pressing technology that was employed during the

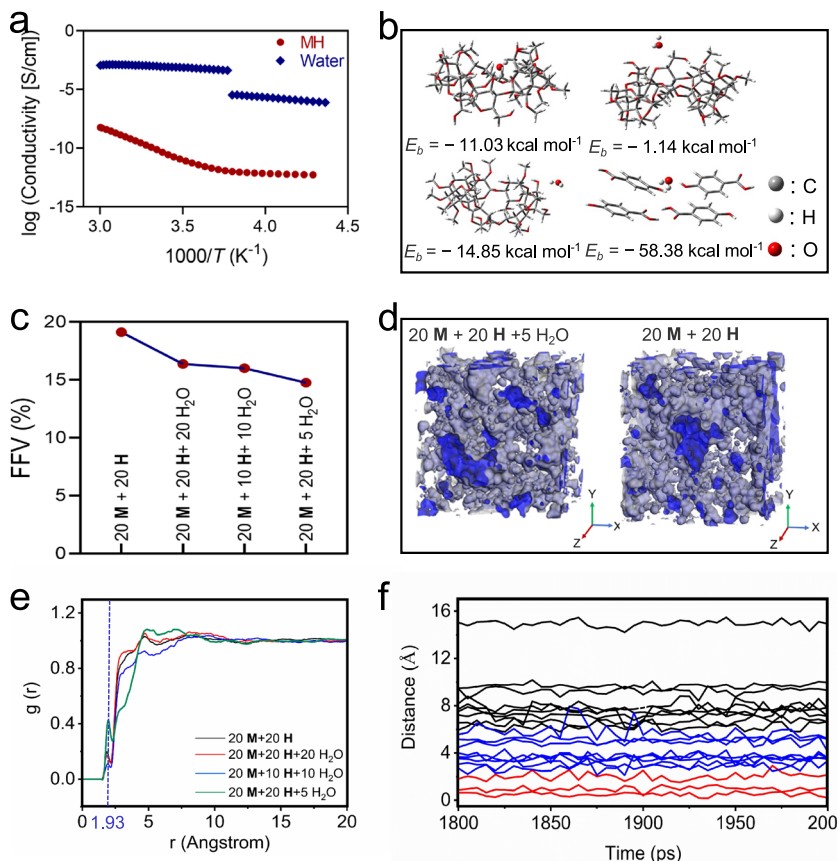

**Fig. 3 | Characterizations and simulations of MH. a** The dependence of DC conductivity $\sigma_{DC}$ versus $1/T$ for **MH** and water. **b** Binding energy and model of **M** and **H** with H$_2$O. **c** FFV of **M** + **H** + H$_2$O. **d** Model of the molecular dynamic of 20**M** + 20**H** + 5H$_2$O and 20**M** + 20**H**. **e** RDF for H atom of **H** and O atom of **M**. **f** The variation of centroid distances of **M/H** with different times (20**M** + 20**H** + 5H$_2$O).

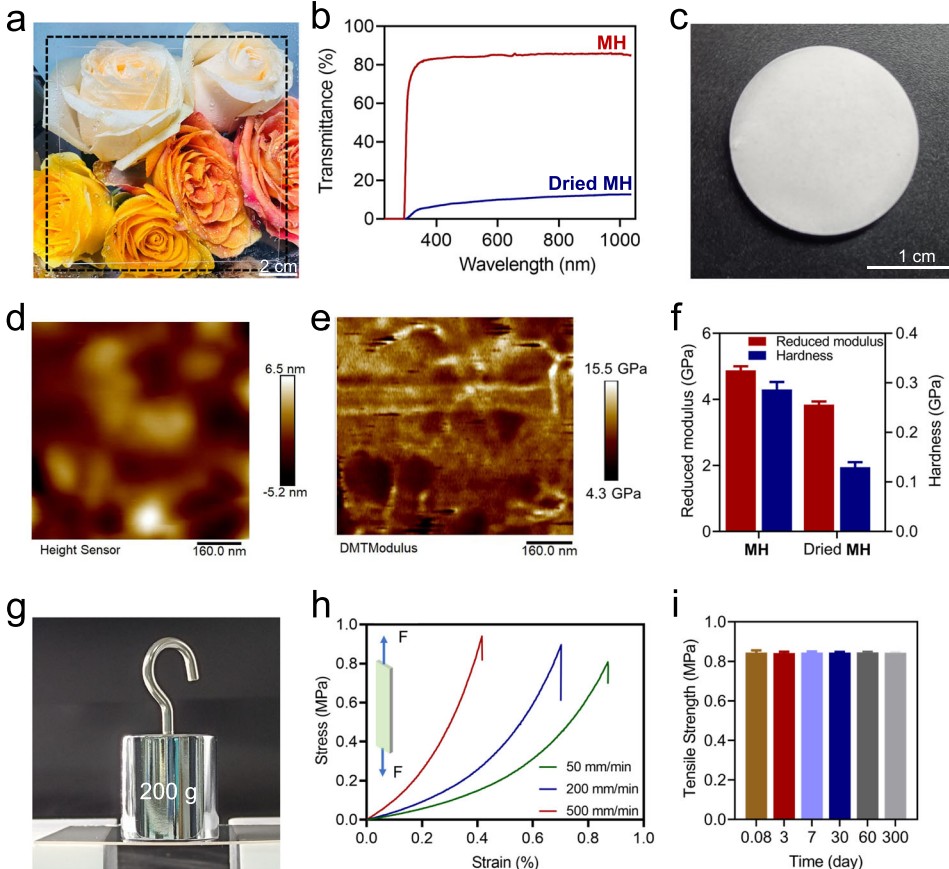

**Fig. 4 | Properties of MH. a** Photo of **MH**. **b** Transmittances of **MH** and vacuum-dried **MH**. **c** Photo of dried **MH**. d. AFM phase image of **MH**. **e** AFM modulus image of **MH**. **f** Reduced modulus and hardness of **MH** and dried **MH** (from nanoindentation). **g** Weight loading (200 g) test of **MH**. **h** Tensile stress-strain curves of **MH**. **i** Time-dependent tensile stress of **MH** at room temperature and 30 RH%. Error bars correspond to the standard deviation of 3 measurements for each analysis.

preparation. The nanoindentation characterizations of **MH** and dried **MH** were carried out (Fig. 4f). The reduced modulus and hardness of **MH** are 4.88 and 0.29 GPa, respectively, which are considerably higher than these of dried **MH** (3.84 and 0.13 GPa), respectively. However, as shown in Figs. 4g, h, **MH** is a relatively fragile glass material from the macroscopic perspective, with a tensile stress strength of 0.85 MPa, respectively, owing to the absence of covalently linked polymeric backbones[40].

Notably, **MH** has good stability and mechanical strengths under various temperature or humidity (Fig. 4i and Supplementary Fig. 22). When the temperatures are kept between 4 and 80 °C, **MH** show tensile strengths between 0.51 and 0.84 MPa. **MH** has the tensile strength of 0.84 MPa when the relative humidity is 5 RH%. According to the TGA data, the initial decomposition temperature of **MH** is above 200 °C. After prolonged standing (6 months), **MH** remains transparent, colorless, and intact, without becoming hygroscopic or dehydrate (TGA results of freshly prepared and long-time stored **MH** samples are nearly the same, in Supplementary Fig. 6). Moreover, the tensile strength does not decrease during the long-term tests (0.85 and 0.84 MPa for 0.08 and 300 days, respectively). No cracks or other flaws are observed on either the surface or the inner part of **MH** by the naked eye. Compared with some supramolecular bulk materials, including glasses and adhesives, **MH** is highly resistant to moisture, further highlighting its advantages[35,36,40].

### Recyclability and compatibility of supramolecular glass
According to the dynamic mechanic thermal analysis, **MH** has a glass transition temperature at 86.10 °C (Supplementary Fig. 23). The relative low glass transition temperature of **MH**, long with its dynamic and

reversible driving forces and good thermal responsiveness make it suitable as a reusable material. **MH** can be recycled through the following routes (Fig. 5a): a) **MH** is dissolved in water, and the aqueous solution is evaporated upon heating; subsequently, the obtained crude product is hot-pressed to yield new **MH**; b) **MH** is cut into small pieces, which are ground into particles with diameters of 2.0−6.0 μm. These particles are placed in a suitable mold and directly hot-pressed to generate a new supramolecular glass. The transmittances of the newly formed **MH** (84.50%, route a, Supplementary Fig. 19) are comparable to that of untreated **MH** glass (85.60%). After multiple cycles (route b), the optical properties of **MH** are not noticeably attenuated, thus highlighting the good recyclability of the supramolecular glass (Fig. 5b). For example, the transmittances of **MH** obtained after the third and fifth cycles at 800 nm are 86.10 and 83.80% (route b), respectively.

A series of inorganic and organic compounds, including $AgNO_3$ ($Ag^+$), $CuSO_4$ ($Cu^{2+}$), $FeCl_3$ ($Fe^{3+}$), $CrCl_3$ ($Cr^{3+}$), $NiCl_2$ ($Ni^{2+}$), $CoCl_2$ ($Co^{2+}$), sudan II (SD), 1,4-bis-(α-cyano-4-methoxystyryl)−2,5-dimethoxybenzene (BDD), and tetrakis(4-hydroxyphenyl)ethylene (AIE), was selected to study their compatibility with **MH**. Water-soluble additives were directly dissolved in **MH** aqueous solution, whereas water-insoluble additives were initially dissolved in ethanol and subsequently mixed with **MH** solution. As shown in Figs. 5c-e and Supplementary Figs. 24 − 26, additives are uniformly dispersed in supramolecular glass. The modified **MH** materials exhibit high transmittances (>84%), as confirmed by quantitative measurements and macroscopic observations (Fig. 5f and Supplementary Fig. 20). Metal cations can effectively improve the microscopic mechanical strength of supramolecular glass (Fig. 5g), owing to the metal coordination

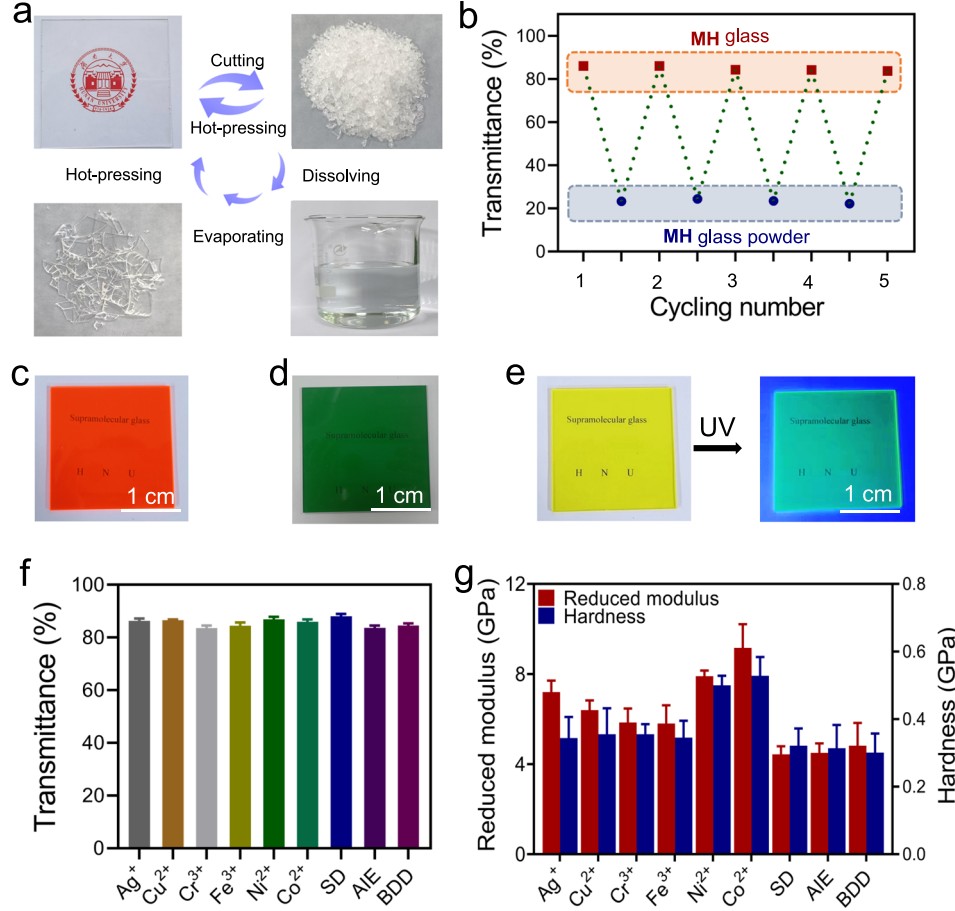

**Fig. 5 | Recyclability and compatibility of MH. a** Recyclability photos of **MH.**
**b** Transmittances of recycled **MH** (800 nm, route b). **c** Photo of **MH** with SD (5 wt%).
**d** Photo of **MH** with CrCl₃ (10 wt%). **e** Photo of **MH** with AIE (1 wt%). **f** Transmittances
of **MH** with additives (800 nm). **g** Reduced modulus and hardness of **MH** with
additives (from nanoindentation). Error bars correspond to the standard deviation
of 3 measurements for each analysis.

effect[55]. Among all the tested metal cations, the mechanical strength of
**MH** is most noticeably enhanced by $Co^{2+}$. The reduced modulus and
hardness of $Co^{2+}$-**MH** are 9.16 and 0.53 GPa, respectively, which are 1.88
and 1.83 times those of unmodified **MH**. Fluorescence emissions of
organic dyes in **MH** were observed (Supplementary Figs. 29 – 33), with
high quantum yields (1,4-bis-[α-cyano-4-methoxystyryl]−2,5-dime-
thoxybenzene 50.73%, tetrakis(4-hydroxyphenyl)ethylene 19.97%).
Meanwhile, **MH** with $Co^{2+}$ or tetrakis(4-hydroxyphenyl)ethylene still
has low roughness and high DMT modulus (Supplementary
Figs. 27,28). These observations indicate that the formation of a glass
structure does not considerably affect the fluorescent behavior of the
additives. On contrast, **MH** shows good compatibility to a variety of
inorganic and organic additives.

## Discussion

In summary, we developed a feasible strategy for constructing supra-
molecular glass, a new type of transparent materials, via the self-
assembly of small organic molecules. The design of supramolecular
glass primarily depends on the molecular recognition behavior
between methyl-β-cyclodextrin [**M**] and *para*-hydroxybenzoic acid [**H**].
The host–guest recognition patterns of **M** and **H** endow **MH** with a
typical intrinsic feature of glass, that is, coexisting short-range order
(i.e., the threaded **M/H** complex) and long-range disorder (polymeric
network) structures. Utilizing host–guest complexation and structural
water, **MH** displays excellent optical behavior, which is comparable to
that of modern glass. Particularly, two significant advantages of **MH**,
namely, recyclability and compatibility, have been successfully
achieved, thereby expanding the applicability of bulk supramolecular

glass. Considering the diversity of macrocycles and host–guest com-
plexes, our study will open up new possibility in the selection of
available recognition motifs for glass formation. The strategy illu-
strated in this study has potential as a universal design concept for
transparent supramolecular materials.

## Methods

### Chemicals, materials, and characterizations

Methyl-β-cyclodextrin (**M**), *para*-hydroxybenzoic acid (**H**),
$CuSO_4 \cdot 5H_2O$, $FeCl_3$, $CrCl_3 \cdot 6H_2O$, $NiCl_2 \cdot 6H_2O$, $CoCl_2 \cdot 6H_2O$, sudan II
(SD), tetrakis(4-hydroxyphenyl)ethylene (AIE), 2,5-dimethoxy ter-
ephthalaldehyde, and 4-methoxyphenylacetonitrile were from
Shanghai Adamas-beta Reagent Co., Ltd. Other commercially solvents
and materials were used directly. Nuclear magnetic resonance (NMR)
spectra were obtained by a Bruker-AV400 MHz spectrometer. Solid-
phase UV spectra were recorded using a Shimadzu UV-3600i Plus.
Infrared (IR) spectra from 4000 to 500 $cm^{-1}$ were tested on a Thermo
Scientific Nicolet iS10 FT-IR spectrometer. Powder X-ray diffraction
(PXRD) was performed on Ultima IV. Thermogravimetric analysis
(TGA) tests were performed using a Shimadzu TGA-50 in nitrogen
atmosphere. Differential Scanning Calorimeter (DSC) was carried out
using a Mettler DSC3 in a nitrogen atmosphere. Mechanical strength
was tested by a universal testing machine (HT-101SC-5). Dynamic
thermomechanical analyses (DMA) were conducted on a DMA 8000-
PerkinElmer using the shear model. Small-angle X-ray scattering
(SAXS) was conducted on an Anton Paar SAXSess. Brunner–Emmet
–Teller (BET) surface area analyses were obtained by a Belsorp-max
ASAP2460. Electron spray ionization (ESI) mass analysis was

performed on a LCMS-IT-TOF. Nanoindentation was obtained by a Hysitron TI 950. Broadband dielectric data was obtained from a Novocontrol Concept 80 system with an Alpha impedance analyzer and a temperature control. To avoid the evaporation of water during the tests, a sealed liquid parallel sample cell BDS 1308 (Novocontrol) was used. Atomic force microscopy (AFM) measurements were conducted on a Bruker Dimension Icon AFM. Transmittance measurements were determined on a Ruike RK-6000 spectrometer. Density measurements were performed on a XingYun XY-2CM. The refractive index was recorded on Horiba UVISEL PLUS. The hot-pressing process was performed on a PCH-600C. Fluorescence spectroscopy and quantum yield were performed on an Edinburgh Instruments FLS980. Low-field nuclear magnetic resonance (LF-NMR) was performed on a MesoMR23-060H-I. Rheology measurements were conducted on an Anton Paar MCR 92.

### Synthesis of 1,4-Bis-(α-cyano-4-methoxystyryl)−2,5-dimethoxybenzene

4-Methoxyphenylacetonitrile (151.6 mg, 1.02 mmol) and 2,5-dimethoxy terephthalaldehyde (100 mg, 0.51 mmol) were mixed in *t*-BuOH (9 mL) and THF (3 mL), which was heated at 50 °C. *t*-BuOK (5.7 mg) and Bu₄NOH (1 mL, 1 M in MeOH) were added quickly and the reaction was stirred for 15 min at 50 °C to form an orange precipitate. The precipitate was separated by filtration, thoroughly cleaned with methanol, and finally dried under vacuum at 50 °C to obtain the target product[56].

### Preparation of supramolecular glass

Briefly, **M** (1.303 g, 0.001 mol), **H** (0.138 g, 0.001 mol) and water (5 mL) were mixed in a beaker, then the mixture was evaporated in an oven at 80 °C for 6 h. The supramolecular glass was cast by hot-pressing for 10 min. The hot-pressing temperature is 80 °C and the pressure is 20 MPa[41,57].

### Measurements of MH mechanical properties

**MH** samples were placed under different conditions (temperature, time, and humidity) for varied times before the measurements.

### Preparation of functionalized MH

**MH** was dissolved in water to form homogeneous solutions, then dyes or fluorescent substances were added to **MH** solutions. The solutions were heated in an oven at 80 °C. The mixtures were hot-pressed at 80 °C, and the pressure is 20 MPa. In some cases, the additives are water-insoluble. Water is replaced by ethanol. When those **MH** sample were prepared, they were placed over colored images for taking photos.

### Preparation of dried MH

Dried **MH** was prepared by vacuuming freshly prepared **MH** at 25 °C and 3 ~ 7 Pa vacuum using a SBC-12 ion sputtering instrument until **MH** is opaque.

### Molecular docking

In this study, Autodock 4.0 was used to conduct molecular docking experiments[58]. **M** was used as a rigid receptor molecule to limit its flexibility during docking[59]. RDkit chemical information software package was used to carry out the three-dimensional transformation of **M** and **H** structures, and MMFF94 force field was used to optimize[60,61]. Furthermore, in order to make docking more accurate, AM1-BCC charge is used for molecular local charge calculation[62]. The docking center coordinates X, Y, and Z are 10.25, 22.47 and 82.28, respectively. The size of the simulated box is set to a cube with a side length of 22.5 Å.

### Molecular dynamic (MD) simulations

MD simulations were carried with Materials Studio (MS) software package[63]. The structures were obtained by the Amorphous Cell module and the random morphology packed in a three-dimensional periodic cell was built by the Monte Carlo algorithm method[64]. Dynamic simulation and geometry optimization of the composite structure are performed with the Forcite module. The force field used in this study is the PCFF[65]. Both MD systems were simulated for 2000 picoseconds with an NPT thermodynamic ensemble at 298.0 K temperature and 0.1 MPa. Fraction of free volume (*FFV*) is calculated as follows Eq. (1).

$$FFV = \frac{V_f}{V_{sp}} = (V_{sp} - 1.3V_w)/V_{sp} \qquad (1)$$

Here $V_{sp}$, and $V_w$ are the cell volume and the van der Waals volume obtained from the van der Waals surface, respectively. $V_f$ is the free volume.

Cohesive energy density was calculated according to the following Eqs. (2−3).

$$CED = \frac{E_{coh}}{V} \qquad (2)$$

$$E_{coh} = -E_{intre} = E_{intra} - E_{total} \qquad (3)$$

Here $E_{coh}$ is the cohesive energy density; $V$ is the volume of a system; $E_{intre}$ is the total energy between all molecules; $E_{intra}$ is the intra molecular energy; $E_{total}$ is the total energy of a system.

The binding energy was calculated according to the following Eq. (4).

$$E_b = E_{total} - E_A - E_B \qquad (4)$$

Here $E_b$, $E_{total}$, $E_A$, and $E_B$ are the binding energy, the total energy, the energy of component A, and the energy of component B, respectively.

## Data availability

The data that support the plots within this paper and other finding of this study are available from the corresponding author upon request.

Crystallographic data for *para*-hydroxybenzoic acid generated in this study have been deposited at the Cambridge Crystallographic Data Center, under deposition numbers CCDC 2239891. Copies of the data can be obtained free of charge via https://www.ccdc.cam.ac.uk/structures/.

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

## Acknowledgements

S.D. acknowledges the National Natural Science Foundation of China (22271087), the Outstanding Youth Scientist Foundation of Hunan Province (2021JJ10010), and the Huxiang Young Talent Program from Hunan Province (2018RS3036). Z.T. acknowledges the Huxiang Young Talent Program from Hunan Province (2021RC3116), the Agricultural Science and Technology Innovation Program (ASTIP-IBFC08), and the earmarked fund for the China Agriculture Research System (CARS-16-E24).

## Author contributions

S.D. and Z.T. supervised the project and designed the experiments. C.C. performed all the experiments and characterization. S.W., Y.Z., and F.L. analyzed the data, and participated in the writing of the manuscript. All authors commented on the manuscript.

## Competing interests

The authors declare no competing interests.
