## [Peer Review File · Nature Communications]

REVIEWER COMMENTS

Reviewer #1 (Remarks to the Author):

This work developed a strategy to construct transparent supramolecular glass by host–guest complexes between methyl- β -cyclodextrin and para-hydroxybenzoic acid as the building blocks. The recyclability, compatibility and thermal processability of this supramolecular glass enabled by dynamic assembly makes it an attractive sustainable material. The authors utilized a range of experimental tools as well as simulations to characterize the structure of this glass, however, many experimental details are missing, and explanations of the experimental results are inadequate. In particular, the role of water involving the supramolecular glass is unclear. The following comments should be addressed, and associated revisions made before the manuscript can be properly assessed.

1. In Figure 1a, the schematic for the mixture of M and H needs to be improved. The role of the dashed line is confusing. If the dashed line represents repeating H unit monomers as a supramolecular polymer, then why does the ratio of H:M look much higher than 1:1? Furthermore, is the formation of the supramolecular polymer from repeating H units independent of the host molecule? This is unclear from the schematic or text.
2. The authors mention that in the control experiments, supramolecular glass was successfully cast from M and H mixtures at molar ratios of 1:2 or 2:1. Were these the only ratios that were tested? Can a glass be formed from each of the components separately?
3. Page 8, the argument on “no order structure is found in the long-range” by showing figure S14-S16 without any description of the experimental result is not appreciated. For example, in Figure S14, the PXRD spectra shows that M and MH presents an amorphous peak at around 10 and 20 degrees while that peak for H is not identified. A more elaborate explanation should be added to this result (as well as to other SI figures), like the peak position/strength and corresponding structure.
4. Figure S16, what is the difference between the red and black curves during the heating process? What is the reason for the endothermic peak at ~ 150 °C? Is it corresponding to some specific structure change (such as crystallization/melting) inside the material?
5. On page 8, “in contrast, after heating under vacuum, MH becomes an opaque material and the visible-light transmittance is below 10%”. The reviewer would like to know more details about the heating condition, including heating time and temperature. Importantly, the water-reducing ratio during the heating process is crucial to illustrate the significance of water in maintaining the high transparency of

HA. As microscopic phase separation and crystallization is concerned in the transparency-opaque transition process, a reliable and quantitative experiment (such as XRD, SAXS, DSC) is needed to characterize the crystallization degree of MH at various heating stage. It would be interesting to show how the transparency of MH involves during the heating condition and corresponding crystallization degree in Figure 4b.

6. Following the above comments, a paper would be more convincing to clarify the water state in a polymer network where XRD with different humidity conditions to sample is applied. "Zhu, J., Andres, C. M., Xu, J., Ramamoorthy, A., Tsotsis, T., & Kotov, N. A. (2012). Pseudonegative thermal expansion and the state of water in graphene oxide layered assemblies. *Acs Nano*, 6(9), 8357-8365."

7. In Page 10, to illustrate the good stability of MH under various conditions, the author claims that low temperature and dry conditions do not affect the optical and mechanical properties of MH. But in the previous discussion, the heating process does reduce the transparency of the MH material which is assumed to the loss of water upon heating. It is unconvincing whether the MH is stable in dry conditions considering water evaporation. Corresponding data needs to be added to support this argument.

Reviewer #2 (Remarks to the Author):

In this paper, the authors prepared supramolecular glass from the dried inclusion complex of methyl- β -cyclodextrin and para-hydroxybenzoic acid by pressing at a higher temperature. The authors characterized the optical, mechanical, thermal, and surface properties of the supramolecular glass. The supramolecular glass was transparent in the visible light region, although Young's modulus of the supramolecular glass was much lower than that of conventional glass. It is noteworthy that the supramolecular glass exhibited good recyclability. Since supramolecular glass is one of the hot topics in materials science, this paper should provide significant insight into this field. Thus, I would recommend publication of this paper after appropriate revision. Specific points to be attended to are listed below.

Major points:

- (1) I am wondering how general this methodology is. The authors should indicate the data for other combinations of cyclodextrins (unmodified and modified) and guests.
- (2) The crystal structure of the M/H inclusion complex should be indicated.
- (3) The glass transition temperature of supramolecular glass is missing.

Minor points:

(4) Page 7, line 11-12. The term “supramolecular polymerization” may be misleading because supramolecular polymers possess defined structures (e.g., linear) formed from monomer units through non-covalent bonding.

(5) Page 7, line 16-19 and Figure 2. (i) Figure 2 does not contain any panels.

(ii) I wonder how the authors evaluated the FFV values. Using the docking results?

(6) Page 8, line 8-10. How did the authors determine the distribution ratios?

(7) Supporting Information, page S3, line 1 from bottom. “Horoba” should be “Horiba”.

(8) Figure S2. The phasing should be improved for the NOESY spectrum.

Reviewer #3 (Remarks to the Author):

Tan, Dong, and coworkers report the design and characterization of transparent supramolecular glasses from host-guest molecular recognition between cyclodextrin and hydroxybenzoic acid. The existence of structural water molecules involved in the stabilization of the self-assembling structure has been demonstrated as well as the compatibility and recyclability of the material. While many of the experiments are well-executed, the innovation and advancements in terms of chemical design and properties of these supramolecular glasses is unclear compared to previously reported supramolecular glass systems. As such, I believe this manuscript requires thorough revision before it will be of sufficient interest to the interdisciplinary readership of Nature Communications. I have outlined specific comments and concerns with the work below:

1/ The manuscript needs to be carefully revised and edited to fix numerous issues regarding the reporting of the figures in both the manuscript and supplementary information within the main text. Almost all figures are incorrectly reported in the main text which makes following the narrative extremely difficult. Some figures simply do not seem to be discussed, and some data are discussed without evidence (e.g., properties of MH after 6 months). Overall, the main text contains many experiments and data with very little explanation and discussion of the reasons for showing these data and the significance of the findings. All sections should be carefully improved to better explain the different experiments, why they were conducted, and how the data and interpretation interact with the main hypotheses and conclusions. Moreover, there are a handful of figures in the supporting information that are not properly discussed in the main text and leave the reader wondering about their impact in the study (e.g., the concentration-dependent NMR spectra of MH in Figure S1).

2/ In Figure 1b, there is a noticeable difference of the shape of the peaks corresponding to the aromatic protons involved in the host-guest interactions compared to free H. This difference could be related to a

modification of the environment of the protons. It could be worth discussing this hypothesis to reinforce the establishment of host-guest interactions.

3/ Did the authors investigate any side reactions occurring during vacuum heating that would change the characteristics of the glass independently from water evaporation?

4/ There is a lack of description of Figures S27-S29. Where do the patterns present in these images come from? More detailed descriptions of the materials and methods, which is required for almost all experiments of the manuscript, should be added to the manuscript to enable readers to potentially reproduce these experiments.

5/ Regarding the experiments in Figures S27-S29, the authors indicate: "Furthermore, we successfully obtained typical fluorescence emissions of organic dyes in MH, indicating that the formation of a glass structure does not considerably affect the fluorescent behavior of the additive". Fluorescence-based experiments of the free dyes and dyes embedded in the glasses should be performed to further reinforce this statement beyond showing these simple images.

A point-by-point response

Reviewer #1:

1. In Figure 1a, the schematic for the mixture of M and H needs to be improved. The role of the dashed line is confusing. If the dashed line represents repeating H unit monomers as a supramolecular polymer, then why does the ratio of H:M look much higher than 1:1? Furthermore, is the formation of the supramolecular polymer from repeating H units independent of the host molecule? This is unclear from the schematic or text.

Response: We are sorry to the confused content in Figure 1a. A new figure was added in the revised manuscript. The newly added Figure 1a was used to describe different building blocks in the glass formation, including methyl- β -cyclodextrin, para-hydroxybenzoic acid, the threaded/unthreaded structures of cyclodextrin and acid, and water.

2. The authors mention that in the control experiments, supramolecular glass was successfully cast from M and H mixtures at molar ratios of 1:2 or 2:1. Were these the only ratios that were tested? Can a glass be formed from each of the components separately?

Response: In the preliminary experiments, **M/H** samples with different molar ratios were carefully studied. It was found that supramolecular glass can be constructed at specific ratios, which were listed in the

following table. Meanwhile, **MH** with the molar ratio at 1:1 shows the best performances in mechanical strength and optical behavior. Unfortunately, only **M** or **H** cannot form supramolecular glass by the same solvent-evaporation strategy.

Table 1. Molar ratio of glass formation

M:H	1:0	10:1	5:1	5:2	2:1	1:1	1:2	1:4	1:8	1:10	0:1
Glass formation	No	No	No	Yes	Yes	Yes	Yes	No	No	No	No

3. Page 8, the argument on “no order structure is found in the long-range” by showing figure S14-S16 without any description of the experimental result is not appreciated. For example, in Figure S14, the PXRD spectra shows that **M** and **MH** presents an amorphous peak at around 10 and 20 degrees while that peak for **H** is not identified. A more elaborate explanation should be added to this result (as well as to other Supplementary Information (SI) figures), like the peak position/strength and corresponding structure.

Response: We are sorry for the unclear description of the state of supramolecular glass. A new description was added in the revised manuscript and supporting information. Meanwhile, the descriptions of other SI Figures were added in the revised supporting information.

As shown in Figure S12, there are only broad peaks between 10 to 30 degree in the PXRD spectrum of supramolecular glass **MH**. This structural

information indicates that **MH** is an amorphous material without obvious crystallization behavior. In contrary, individual acid is crystalline. However, after noncovalent molecular recognition and polymerization, the stacking tendency of acid is strongly suppressed. This transition can be ascribed to the following reasons: the hydrogen bonding formation between cyclodextrin and acid inhibit the ordered aggregation of acid; during the solvent evaporation process, there is a rapid transition from a solution to a solid, during which a fast decrease of viscosity happens. Thus acid can not return to a more stable crystal state. Meanwhile, in our previous studies, such transitions from crystalline compounds to amorphous materials has been observed. (Mater. Horiz., 2023,10, 5152-5160; Angew. Chem. Int. Ed. 2020, 59, 11871).

4. Figure S16, what is the difference between the red and black curves during the heating process? What is the reason for the endothermic peak at ~150 °C? Is it corresponding to some specific structure change (such as crystallization/melting) inside the material?

Response: We added related descriptions in the revised supporting information.

We used a relatively complicated DSC measurements for two main reasons: a) there are water molecules located in the cavity of cyclodextrin: b) because water evaporation method was used, **MH** contains water

molecules. Therefore, the role of water should be considered in DSC measurements.

First, the glass sample was heated from $-80\text{ }^{\circ}\text{C}$ to $200\text{ }^{\circ}\text{C}$ (the red curve). Then, the glass sample was cooled down from 200 to $-80\text{ }^{\circ}\text{C}$ (the blue curve). Finally, the glass sample was heated from -80 to $200\text{ }^{\circ}\text{C}$ (the black curve).

The broad peak at around $75\text{ }^{\circ}\text{C}$ can be ascribed to the loss of water molecules located in the cavity of cyclodextrin. The endothermic peak appearing at $150\text{ }^{\circ}\text{C}$ is caused by the loss of bound water. This is also consistent with our TGA data. Additionally, according to the PXRD results, no crystallization was observed. DSC measurement did not support the melting behavior of MH at $150\text{ }^{\circ}\text{C}$. Meanwhile, after heating **MH** at $150\text{ }^{\circ}\text{C}$, we did not observe any melting phenomenon on **MH** by naked eye.

5. On page 8, “in contrast, after heating under vacuum, MH becomes an opaque material and the visible-light transmittance is below 10%”. The reviewer would like to know more details about the heating condition, including heating time and temperature. Importantly, the water-reducing ratio during the heating process is crucial to illustrate the significance of water in maintaining the high transparency of HA. As microscopic phase separation and crystallization is concerned in the transparency-opaque transition process, a reliable and quantitative experiment (such as XRD,

SAXS, DSC) is needed to characterize the crystallization degree of MH at various heating stage. It would be interesting to show how the transparency of MH involves during the heating condition and corresponding crystallization degree in Figure 4b.

Response: A detailed description of the dried **MH** was added in the revised manuscript. Dried **MH** was prepared by vacuuming at 25 °C and 3~7 Pa vacuum using a SBC-12 ion sputtering instrument. Heating is not needed during the preparation of dried **MH**. We are sorry for the wrong description “heating”. During the drying process, a transition of **MH** from transparency to translucence, finally to opacity was clearly observed by naked eyes and quantitatively measurements (Figure is listed below).

Dried **MH** was carefully characterized by SASX, XRD, and DSC. It was found that the dry material is still an amorphous material with the T_g at ca. 135 ° C. These observations demonstrate that the removal of water does not destroy the microscopic structure of **MH**, which can be attributed to the strong supramolecular interactions between cyclodextrin and acid at room temperature.

It is obvious that **MH** without water is an opaque material. This transition can be ascribed to the following reasons: 1) the evaporation of water molecules in **MH** leads to the formation of structures full of air; 2) those newly formed structures have obvious light scattering behavior.

The role of water was carefully studied and more experiments were carried

out. From the dielectric spectra, no discontinuity was observed in the plots of σ_{DC} over $1/T$ for **MH**, indicating that the overwhelming majority of water molecules in **MH** does not exist as bulk water cluster. Low-field NMR spectrum demonstrate that around 98.4% of water is bound water. In our previous study (Sci. Adv. 2017, 3, eaao0900), it has been found that a small number of water molecules serve as essential monomers in supramolecular polymerization. These water molecules are referred as “structural water” to distinguish them from common solvent water. In this work, water molecules are important monomers in supramolecular polymerization and can increase the cross-linking density of **MH**. According to reported references and TGA results of **MH**, water molecules in **MH** can be mainly divided into two classes: cyclodextrin can complex water molecules in its cavity; water molecules can bridge with **M** and **H** via hydrogen bonds (J. Phys. Chem. B 2021, 125, 11112–11121; Sci. Adv. 2017, 3, eaao0900).

Figure 1. Transmittances of **MH** vacuuming for different vacuum time.

Figure 2. PXRD of dried MH.

Figure 3. SAXS of dried MH.

Figure 4. DSC of dried MH.

Figure 5. The dependence of DC conductivity σ_{dc} versus $1/T$ for **MH** and water.

6. Following the above comments, a paper would be more convincing to clarify the water state in a polymer network where XRD with different humidity conditions to sample is applied. “Zhu, J., Andres, C. M., Xu, J., Ramamoorthy, A., Tsotsis, T., & Kotov, N. A. (2012). Pseudonegative thermal expansion and the state of water in graphene oxide layered assemblies. *ACS Nano*, 6(9), 8357-8365.”

Response: We really appreciated the comment. The related *ACS Nano* paper was cited in the revised manuscript. Meanwhile, a section of the role of water in supramolecular glass was listed in the maintext.

PXRD results of **MH** with different water contents has been added to Figure S13 in the revised supporting information. Those results indicate that the water content does not exert great influences on the amorphous state of supramolecular glass, and it was failed to study the role of water in

MH by PXRD. Therefore, dielectric and low-field NMR experiments were carried out to understand the water molecules in **MH** (in the section of “Structural water in supramolecular glass”). From the dielectric spectra, no discontinuity was observed in the plots of σ_{DC} over $1/T$ for **MH**, indicating that the overwhelming majority of water molecules in **MH** do not exist as bulk water clusters (free water). Low-field NMR spectrum demonstrate that around 98.4% of water is bound water (not the free water). These water molecules are referred as “structural water” to distinguish them from common solvent water.

Figure 6. PXRD of different water contents MH.

Figure 7. Low field nuclear magnetic resonance of MH.

7. In Page 10, to illustrate the good stability of MH under various conditions, the author claims that low temperature and dry conditions do not affect the optical and mechanical properties of MH. But in the previous discussion, the heating process does reduce the transparency of the MH material which is assumed to the loss of water upon heating. It is unconvincing whether the MH is stable in dry conditions considering water evaporation. Corresponding data needs to be added to support this argument.

Response: We are sorry for the confused description of the stability. New description was added in the revised manuscript.

According to the TGA experiments and long-term tests, it was found that water molecules in **MH** are stable in the dry and low temperature (**MH** is still transparent), because water molecules in this study can be recognized as “structural water”. In our previous study, it has been found that “structural water” has better stability, compared with free water. Structural water is hardly removed at room temperature and/or low humidity.

There are two methods to remove water from **MH**, heating or evaporation under vacuum. The loss of water was not observed at room temperature or at low humidity. Time-dependent TGA experiments clearly showed the good stability of **MH**, because the water loss curves of freshly prepared and long-time stored **MH** samples are almost the same. The decomposition temperature of **MH** is ca. 300 °C.

Figure 8. TGA of MH.

Reviewer #2:

In this paper, the authors prepared supramolecular glass from the dried inclusion complex of methyl- β -cyclodextrin and para-hydroxybenzoic acid by pressing at a higher temperature. The authors characterized the optical, mechanical, thermal, and surface properties of the supramolecular glass. The supramolecular glass was transparent in the visible light region, although Young's modulus of the supramolecular glass was much lower than that of conventional glass. It is noteworthy that the supramolecular glass exhibited good recyclability. Since supramolecular glass is one of the hot topics in materials science, this paper should provide significant insight into this field. Thus, I would recommend publication of this paper after appropriate revision. Specific points to be attended to are listed below.

Major points:

(1) I am wondering how general this methodology is. The authors should

indicate the data for other combinations of cyclodextrins (unmodified and modified) and guests.

Response: We really appreciated this comment.

More experiments were carried to study the universality of this methodology. According to the macroscopic phenomena, it was found that β -cyclodextrin cannot form glass with *p*-hydroxybenzoic acid. While methyl- β -cyclodextrin can form glass materials with a wide series of acids, which was listed in the follow table. After testing the mechanical and optical properties of formed glasses, it was found that supramolecular glass from *p*-hydroxybenzoic acid and methyl- β -cyclodextrin has the best mechanical and optical performances. Therefore, **MH** was selected in this study.

Table 2. Hosts and guests of glass formation

	Host	Guest	Molar ratio	Glass formation
1	Mthyl- β -cyclodextrin	Benzoic acid	1:1	Yes
2	Mthyl- β -cyclodextrin	3-Hydroxybenzoic acid	1:1	Yes
3	Mthyl- β -cyclodextrin	3,4-Dihydroxybenzoic acid	1:1	Yes
4	Mthyl- β -cyclodextrin	2,5-Dihydroxybenzoic acid	1:1	Yes
5	Mthyl- β -cyclodextrin	3,5-Dihydroxybenzoic acid	1:1	Yes
6	Mthyl- β -cyclodextrin	Gallic acid	1:1	Yes
7	Mthyl- β -cyclodextrin	2,3,4-Trihydroxybenzoic acid	1:1	Yes

8	α -cyclodextrin	para -hydroxybenzoic acid	1:1	No
9	β -cyclodextrin	para -hydroxybenzoic acid	1:1	No
10	γ -cyclodextrin	para -hydroxybenzoic acid	1:1	No
11	Hydroxypropyl- β -cyclodextrin	para -hydroxybenzoic acid	1:1	Yes

(2) The crystal structure of the M/H inclusion complex should be indicated.

Response: We have tried our best to obtain the single crystal of **MH** complex. Unfortunately, we failed to get any suitable crystal of **MH**. Thus, we applied simulation to display the possible **M-H** recognition motifs. Meanwhile, we successfully obtained the single crystal of **H**, in which hydrogen bonding was observed.

(3) The glass transition temperature of supramolecular glass is missing.

Response: We were sorry for the missed glass transition temperature of **MH**. The glass transition temperature has been added in the revised supporting information (Table S1).

Minor points:

(4) Page 7, line 11-12. The term “supramolecular polymerization” may be misleading because supramolecular polymers possess defined structures (e.g., linear) formed from monomer units through non-covalent bonding.

Response: This section has been modified.

(5) Page 7, line 16-19 and Figure 2. (i) Figure 2 does not contain any panels.

Response: This section has been modified.

(ii) I wonder how the authors evaluated the FFV values. Using the docking results?

Response: FFV is calculated through molecular dynamics. The detail method is shown in method Molecular dynamic (MD) simulations.

Fraction of free volume (*FFV*) is calculated as follows eq. (1).

$$FFV = \frac{V_f}{V_{sp}} = (V_{sp} - 1.3V_w)/V_{sp} \quad (1)$$

Where V_{sp} is the cell volume and V_w is the van der Waals volume obtained from the van der Waals surface and V_f is free volume.

(6) Page 8, line 8-10. How did the authors determine the distribution ratios?

Response: The distribution ratios used in this study was obtained from molecular dynamics simulation. Four typical **M@H** units with different centroid distances were used to describe the complexation states, as shown in Figure 2a. Based on the centroid distances and related conformation, the distribution ratios targeted on different **MH** complexes were determined. The centroid distance less than 2 Å indicates the completely enclosed structure; the centroid distance between 2 and 6 Å means the semi enclosed structures; the centroid distance greater than 6 Å indicate **M** and **H** are far

away from each other.

(7) Supporting Information, page S3, line 1 from bottom. “Horoba” should be “Horiba”.

Response: We were sorry for this mistake. It has been corrected in the revised supporting information.

(8) Figure S2. The phasing should be improved for the NOESY spectrum.

Response: The NOESY spectra was replaced by other NMR results in revised supporting information.

Reviewer #3:

1. The manuscript needs to be carefully revised and edited to fix numerous issues regarding the reporting of the figures in both the manuscript and supplementary information within the main text. Almost all Figures are incorrectly reported in the main text which makes following the narrative extremely difficult. Some figures simply do not seem to be discussed, and some data are discussed without evidence (e.g., properties of MH after 6 months). Overall, the main text contains many experiments and data with very little explanation and discussion of the reasons for showing these data and the significance of the findings. All sections should be carefully improved to better explain the different experiments, why they were

conducted, and how the data and interpretation interact with the main hypotheses and conclusions. Moreover, there are a handful of figures in the supporting information that are not properly discussed in the main text and leave the reader wondering about their impact in the study (e.g., the concentration-dependent NMR spectra of MH in Figure S1).

Response: We really appreciated the comments from Reviewer 3, which can help us to thoroughly improve our manuscript. The whole manuscript and supporting information were carefully revised. Especially, figures in maintext and supporting information were described in detail. Some results were deleted, according to the description in maintext. All changes were marked with yellow.

2. In Figure 1b, there is a noticeable difference of the shape of the peaks corresponding to the aromatic protons involved in the host-guest interactions compared to free H. This difference could be related to a modification of the environment of the protons. It could be worth discussing this hypothesis to reinforce the establishment of host-guest interactions.

Response: The description of the NMR spectra was added in the revised manuscript and supporting information.

3. Did the authors investigate any side reactions occurring during vacuum

heating that would change the characteristics of the glass independently from water evaporation?

Response: We were sorry for the wrong description of water evaporation process. Water was removed at 25 °C under vacuum, that is to say, no heating was used. By comparing the NMR spectra of **MH** and dried **MH**, it was observed that there are no covalent reactions between **M** and **H** during the water evaporation. Because neither new peaks nor obvious chemical shifts were found in their ^1H NMR spectra.

Figure 9. ^1H NMR spectra of freshly prepared MH and dried MH (400 MHz, D_2O , 25 °C).

4. There is a lack of description of Figures S27-S29. Where do the patterns present in these images come from? More detailed descriptions of the materials and methods, which is required for almost all experiments of the manuscript, should be added to the manuscript to enable readers to

potentially reproduce these experiments.

Response: The related experimental methods were added in the revised manuscript and supporting information.

5. Regarding the experiments in Figures S27-S29, the authors indicate: “Furthermore, we successfully obtained typical fluorescence emissions of organic dyes in MH, indicating that the formation of a glass structure does not considerably affect the fluorescent behavior of the additive”. Fluorescence-based experiments of the free dyes and dyes embedded in the glasses should be performed to further reinforce this statement beyond showing these simple images.

Response: Fluorescence spectroscopy and quantum yield of the free dyes and modified glasses have been studied. It was found that modified glass materials still have good quantum yields, as shown in the following figure.

Figure 10. Quantum yields of MH with dyes. (a) MH with 1,4-bis-(α -

**cyano-4-methoxystyryl)-2,5-dimethoxybenzene; (b) MH with
tetrakis(4-hydroxyphenyl)ethylene.**

REVIEWER COMMENTS

Reviewer #1 (Remarks to the Author):

The authors have diligently conducted the required experiments and implemented substantial revisions in response to my comments and suggestions. Consequently, the manuscript's readability has markedly improved. I am pleased to recommend the publication of the work entitled "Bulk Transparent Supramolecular Glass Enabled by Host–Guest Molecular Recognition" in Nature Communications.

Reviewer #2 (Remarks to the Author):

I have found that the authors revised and improved the manuscript considering the questions I raised previously. Thus, I would recommend publication of this paper after minor revision. The following is an additional comment concerning the comment (1) of previous major points.

- The authors should discuss whether the glass state is thermodynamically stable or kinetically frozen.

Reviewer #3 (Remarks to the Author):

The authors have made great efforts in responding to the comments and concerns of the reviewers, and clarified or added data as necessary. I now recommend publication.

In the revised manuscript and supporting information, we have marked all changes and the related contents in yellow to make them easy to identify. A point-by-point response was prepared. Particularly, the innovation and advancements of supramolecular glass **MH** were imbedded into different sections of the revised manuscript; the description of the role and importance of water was described in the section “Structural water in supramolecular glass” and other parts of the revised manuscript.

A point-by-point response.

1. *In particular, as requested by Reviewer #3 the innovation and advancements in terms of chemical design and properties of this supramolecular glasses in comparison to reported systems that also employed host-guest interactions must be discussed and the discussion implemented in the manuscript file.*

Response: We are sorry for the unclear and inadequate description of the innovation and advancements in the last revision round. The key descriptions and related contents of the advantages of **MH** were imbedded into different sections of this manuscript.

Compared with reported supramolecular glass systems, **MH** glass shows a variety of innovation and advancements in the items of the intrinsic structures, molecular structures, driving force, optical behavior, the role of water, and mechanical properties.

1. Intuitively, glass is a kind of bulk materials with good transparency in visible region. However, the nature and intrinsic structure of glass are complicated and controversial. Usually, glass is defined as a metastable matter with long-range disorder, short-range order (these are the unique characteristics that distinguish glass from other transparent materials, including minerals, resins, and gels). In the reported supramolecular glasses, the studies on the nature and intrinsic of glass were not involved. In this work, we carefully studied the nature and intrinsic structure of glass *via* a combined experimental and theoretical investigation. The coexisted long-range disorder and short-range order structures were identified from the metastable **MH**, indicating that **MH** is a glass material form low-molecular-weight monomers that are bonded through noncovalent interactions.

2. In the reported glass systems, the building blocks are relatively limited, with hydrogen bonded/metal coordinated motifs as the main units. In this study, for the first time, macrocycles and threaded/semi-threaded structures were used to fabricate artificial glass. Considering the diversity of macrocycles and the related threaded structures, our study will open up new possibility in the selection of available recognition motifs for glass formation.

3. In the reported studies, researches were focused on the optical properties (such as colors and fluorescence) or biological degradations. Detailed

studies of their mechanical strengths, hardness, and stability were not thoroughly performed. Additionally, from the limited information from those references, it is indicated that only glass materials in small scales are available. On contrast, supramolecular glass **MH** used in our study not only has novel monomer structures, but also exhibits tough mechanical strength, high hardness, long-term stability, and good compatibility. In addition, **MH** with large sizes (> 10.0 cm) or thin thickness (< 1.0 mm) has been successfully prepared.

4. Most of supramolecular glasses were obtained by the solvent-evaporation methods, while the existence and the importance of water molecules were frequently ignored. The role of water in glass formation is still ambiguous. For the first time, the relationship between optical transparency and water molecules were investigated and discussed in our revised maintext. Our study clearly shows the great importance of water in realizing and maintaining optical transparency (the role of water is described in the section “*Structural water in supramolecular glass*” in the revised maintext).

Besides the above mentioned issues, it was observed that **MH** is not hygroscopic and highly resistant to moisture. Meanwhile, **MH** is stable and transparent in the long-term experiment, and no dehydration phenomenon was observed.

According to the above description and comparison, it is demonstrated

that this work provides a new strategy for the constructing transparent materials from supramolecular recognition motifs.

2. *Additionally, the role of water in the amorphous state of the supramolecular gel must be explained in detail as pointed by Reviewer #1.*

Response: We are sorry for the inadequate description of the role of water in glass formation in the last revision round. In the revised manuscript, an independent section “*Structural water in supramolecular glass*” was added. In this section, the role and importance of water in the glass formation were carefully investigated. Meanwhile, the influence of water in the mechanical strength and transparency were discussed in the section “*Properties of supramolecular glass*”. What is more, the involvement of water in the recycling of **MH** was discussed in the section “*Recyclability and compatibility of supramolecular glass*”. The changes and related contents in the revised maintext are highlighted in the color of yellow in the revised maintext and supporting information.

In this study, according to the thermogravimetric analysis (TGA) result of **MH**, it was observed that supramolecular glass contains approximately 4.0 wt% water (**Fig. 1a**). Broadband dielectric spectroscopy experiments were performed to investigate the water molecules in **MH**. As shown in **Fig. 1b**, no discontinuity is observed in the plots of σ_{DC} over $1/T$ for **MH** around 0 °C (the freeze point of water in bulk), indicating that water

molecules in **MH** may exist in an inseparable state, and the formation of bulk water clusters in supramolecular glass is unfavorable.

Fig. 1. Characterizations and simulations of **MH**. a. TGA spectra of **MH** at different times. b. The dependence of DC conductivity σ_{dc} versus $1/T$ for **MH** and water. c. Low field nuclear magnetic resonance of **MH**. d. Binding energy and model of **M** and **H** with H_2O . e. FFV of **MH**. f. Model of molecular dynamic of **MH**.

The simulated results also show that **M** and **H** can be bonded with water molecules through flexible binding patterns (**Fig. 1d**). Most of the

simulated **M**/water or **H**/water structures have moderate to high binding affinities, with the binding energies from -1.14 to -58.38 kcal mol $^{-1}$. Low-field NMR spectrum provides quantitative results and further supports the existence of structural water, because around 98.4% of water in **MH** is bound water with the relaxation time of 0.1 ms (**Fig. 1c**).

Table 1. Cohesive energy density (J m $^{-3}$).

System	Total	Van der	Electrostatic	Other
20 M +20 H	4.70×10^8	3.34×10^8	1.24×10^8	1.24×10^7
20 M +20 H +20 H ₂ O	5.20×10^8	3.51×10^8	1.56×10^8	1.30×10^7
20 M +10 H +10 H ₂ O	5.01×10^8	3.51×10^8	1.37×10^8	1.33×10^7
20 M +20 H +5 H ₂ O	5.14×10^8	3.72×10^8	1.28×10^8	1.36×10^7

The above results demonstrate that the structural water molecules in **MH** facilitate the glass formation: different **M/H** recognition motifs can self-assemble into three-dimensional networks; the water molecules involved provide additional hydrogen bonding sites and increase the cross-linking density of supramolecular glass. The water molecules effectively increase the cohesive energy density of **MH** from 4.70×10^8 to 5.20×10^8 J m $^{-3}$, indicating that structural water contributes to a higher intrinsic interaction

intensity (Table 1).

Water significantly influences the structural configuration of **MH**. In general, water molecules reduce the free volume of supramolecular glass (Fig. 1e,f). For example, the fraction of free volume (FFV) of water-free **MH** (**M:H** at 20:20) is 19.13%, which is higher than that of **MH** with a molar ratio of **M: H: water** of 20:20:5 (14.75%). Meanwhile, compared with “free water”, structural water is more stable in long-term test (Fig. 1a), because the TGA results of freshly prepared **MH** is almost the same to that of long-term stored **MH** (300 days).

Fig. 2. Properties of **MH**. a. Photo of **MH**. b. Transmittances of **MH** and vacuum-dried **MH**. c. Photo of dried **MH**. d. Reduced modulus and hardness of **MH** and dried **MH** (from nanoindentation).

After drying under vacuum, **MH** becomes an opaque material, and its visible-light transmittance is below 10%, demonstrating that water is important to the good transparency of **MH** (**Fig. 2a-c**). A possible explanation of the transition from transparency to opacity of **MH** is that structural water is replaced by air, which leads to the light scattering performance. In addition, water can improve the reduced modulus and hardness of **MH** (**Fig. 2d**).

Reviewer #1 (Remarks to the Author):

The authors have diligently conducted the required experiments and implemented substantial revisions in response to my comments and suggestions. Consequently, the manuscript's readability has markedly improved. I am pleased to recommend the publication of the work entitled “Bulk Transparent Supramolecular Glass Enabled by Host–Guest Molecular Recognition” in Nature Communications.

Response: We thank the reviewer for the valuable comments that significantly improve our revised manuscript.

Reviewer #2:

1. I have found that the authors revised and improved the manuscript considering the questions I raised previously. Thus, I would recommend publication of this paper after minor revision.

Response: We thank the reviewer for the valuable comments that significantly improve our revised manuscript.

2. The following is an additional comment concerning the comment (1) of previous major points. The authors should discuss whether the glass state is thermodynamically stable or kinetically frozen.

Response: The state of the glass was described in the revised maintext.

The inner structure of glass is closely related to its formation process. In this study, we applied a solvent evaporation strategy, which involves a transition from a diluted solution to a solid glass, during which water molecules were removed and the viscosity of glass increased rapidly (**Fig.3**). Those observations indicate that glass is in a thermodynamically metastable, and the dynamical recognition motifs are frozen, due to the high viscosity.

Fig. 3. Temperature-dependent viscosity of **MH** (V_0 is the viscosity of **MH** at 30 °C; V is the viscosity of **MH** at different temperatures).

Reviewer #3:

The authors have made great efforts in responding to the comments and concerns of the reviewers, and clarified or added data as necessary. I now recommend publication.

Response: We thank the reviewer for the valuable comments that significantly improve our revised manuscript.

REVIEWERS' COMMENTS

Reviewer #3 (Remarks to the Author):

The authors have adequately addressed all of my comments and concerns. I support publication of this work in Nature Communications.